# Semi-Supervised Few-Shot Learning with a Controlled Degree of Task-Adaptive Conditioning

## Abstract

Few-shot learning aims to handle previously unseen tasks using only a small amount of new training data. In preparing (or meta-training) a few-shot learner, however, massive labeled data are necessary. In the real world, unfortunately, labeled data are expensive and/or scarce. In this work, we propose a few-shot learner that can work well under the semi-supervised setting where a large portion of training data is unlabeled. Our method employs explicit task-conditioning in which unlabeled sample clustering for the current task takes place in a new projection space different from the embedding feature space. The conditioned clustering space is linearly constructed so as to quickly close the gap between the class centroids for the current task and the independent per-class reference vectors meta-trained across tasks. In a more general setting, our method introduces a concept of controlling the degree of task-conditioning for meta-learning: the amount of task-conditioning varies with the number of repetitive updates for the clustering space. Extensive simulation results based on the *mini*ImageNet and *tiered*ImageNet datasets show state-of-the-art semi-supervised few-shot classification performance of the proposed method. Simulation results also indicate that the proposed task-adaptive clustering shows graceful degradation with a growing number of distractor samples, i.e., unlabeled samples coming from outside the candidate classes.

## 1 Introduction

Deep neural networks rely critically on massive training with large amounts of annotated data. When training data is scarce and/or unlabeled, which is typically the case in practice, current machine learning algorithms often struggle, failing to optimize deep neural networks. On the other hand, humans can quickly learn new concepts even from noisy and limited experiences. Training machines to learn new tasks rapidly from a small amount of labeled data samples continues to be a daunting challenge in realizing human-like adaptability in machines.

In tackling a related problem in computer vision, many researchers have worked on few-shot learning algorithms, which aim to generalize models for classifying novel classes using only a small number of example images. A popular way of developing few-shot learners is to apply ample initial learning or meta-training in *episodic* form (Vinyals et al., 2016), where the learner is exposed to a large number of widely varying tasks one by one, each time with just a few labeled samples. Built upon the strategy of episodic training and the concept of *meta-learning* or learning to learn, successful few-shot learners strive to achieve both proper inductive bias and task-specific adaptability.

Some methods rely on meta-training the base model without explicit task-dependent conditioning at few-shot-based evaluation time (Koch et al., 2015; Vinyals et al., 2016; Snell et al., 2017; Mishra et al., 2018; Sung et al., 2018; Allen et al., 2019). Of these, Prototypical Networks of (Snell et al., 2017) train a single embedder such that its per-class averages of the features act as *prototypes* for representing given tasks. Despite its simplicity, this method has consistently produced relatively strong few-shot learning results. Some other algorithms try to build a good inductive bias with no fine-tuning at evaluation time, while also employing additional networks on top of the base feature extractor. Temporal convolutional networks with soft attention (Mishra et al., 2018) and relation modules (Sung et al., 2018) are well-known examples. On the other side of the spectrum, there are

algorithms that rely heavily on task-specific adaptation via explicit learning of model parameters at deployment (Finn et al., 2017; Ravi & Larochelle, 2017). These approaches first build a meta-trained initialization model and then execute a number of model parameter updates given a new task. There also exist few-shot learners that employ explicit task-conditioning without direct model parameter updates at evaluation time (Oreshkin et al., 2018; Li et al., 2019; Liu et al., 2019; Yoon et al., 2019). For Transductive Propagation Networks (TPNs) of (Liu et al., 2019), a graph construction module is trained across tasks by repetitively forming an episode-specific graph. On the task-dependent graph, which shows relationship between embedded samples, labels from support samples propagate to query samples for predicting query labels. In (Oreshkin et al., 2018), task-conditioning is done through scaling and shifts of feature vectors at individual layers of the embedding network. More recently, explicit task-conditioning in (Yoon et al., 2019) takes the form of task-adaptive projection networks (TapNets), where new classification space is linearly constructed by quickly zeroing the gap between the embedded output of the current task and the independent reference vectors learned across tasks during meta-training. In yet another type of task-adaptation (Li et al., 2019), a model learns to generate the parameters of a task-specific matching module which processes and compares embedded support and query samples.

While recent advances in few-shot learning algorithms have enabled steady performance improvements, they invariably rely on heavy meta-training with diverse and massive labeled sample images, which is not practical. Semi-supervised few-shot learning algorithms tackle this challenge by allowing a learner to be trained using largely unlabeled data sets. The goal of semi-supervised few-shot learning is to acquire generalization capability by effectively utilizing abundantly available unlabeled data. In an early attempt (Ren et al., 2018), using the base Prototypical Networks, semi-supervised clustering is proposed for leveraging unlabeled examples. There, the per-class prototypes, the class centroids of the embedded features, are first obtained only with labeled samples, after which clusters are formed using unlabeled samples in the embedded space around these prototypes. Based on the distances of each unlabeled sample to the prototypes, soft labeling is done. The original prototypes are then refined by the soft labels, and in turn used for classifying the queries. The key idea of the approach is that the embedding space is meta-trained for both clustering of unlabeled samples to adjust the prototypes and classifying the query samples around them. In a recent work of (Sun et al., 2019), a model learns to self-train a few-shot learner using unlabeled samples. There, unlabeled samples with high confidence scores are cherry-picked, and the self-training weights of the selected unlabeled samples are computed by a meta-trained module. Then, the learner is self-trained through fine-tuning using the selected samples and computed weights.

We also focus on semi-supervised few-shot learning but our approach offers unique task-conditioning algorithms to utilize unlabeled samples that can work well under crucial label-deficient settings. While we utilize TapNets of (Yoon et al., 2019) as our base architecture, we propose a novel way of constructing an alternative *task-adaptive clustering* (TAC) space where the embedded features of unlabeled samples are further projected and clustered. In our TAC method, the projection space where clustering occurs is reconstructed *repetitively* and is in general different from the final classification space. This repetitive projection, given a new episode, also allows controlling the degree of task-conditioning during the evaluation phase. The gain we achieve using our method goes well beyond what is possible with a simple application of baseline TapNets.

Extensive evaluation based on partially labeled *mini*ImageNet and *tiered*ImageNet datasets show that our TAC algorithm achieves best accuracies with considerable margins, with or without the unlabeled distractor samples. We also suggest a more realistic and challenging test environment with a growing portion of distractor samples in every episode. In our test setup, the number of unlabeled samples per class is not uniform as well, which is more reflective of real world settings. Our TAC algorithm exhibits graceful degradation with a growing portion of distractor samples when compared with the simpler Prototypical Networks (Ren et al., 2018).

## 2    TASK-ADAPTIVE CLUSTERING FOR SEMI-SUPERVISED FEW-SHOT LEARNING

### 2.1    PROBLEM DEFINITION FOR SEMI-SUPERVISED FEW-SHOT LEARNING

Episodic training, often employed in meta-training for few-shot learners, feeds the model with one episode at a time, with each episode composed of a support set with a few labeled samples and a query set with samples to be classified (Vinyals et al., 2016). For $N$-way, $K$-shot learning, $N$ classes are first chosen from a training set, and then a support set is formed by taking $K$ labeled samples per chosen class. The query set is also constructed from the samples of these $N$ classes, disjointly with the support set. For every episode, the model parameters are updated by the loss incurred during prediction of the labels for queries. The meta-trained few-shot learners are evaluated on unseen classes from the test set, which is disjoint with the train set.

For semi-supervised few-shot learning, we follow the settings of (Ren et al., 2018). Each episode is constructed by support set $\mathcal{S}$, query set $\mathcal{Q}$, and unlabeled set $\mathcal{U}$, which contains inputs without labels: $\mathcal{U} = \{\tilde{\mathbf{x}}_1, \cdots, \tilde{\mathbf{x}}_U\}$. For $N$-way, $K$-shot semi-supervised few-shot classification, the support and query sets of every episode are composed of images chosen from the $N$ classes while the unlabeled set can contain images coming from irrelevant classes outside the $N$ candidate classes (*distractors*). The unlabeled set $\mathcal{U}$ is also utilized for few-shot classification together with the support set $\mathcal{S}$ for classifying queries in $\mathcal{Q}$.

### 2.2    PRELIMINARIES ON TAPNET

TapNet of Yoon et al. (2019) is a few-shot learning algorithm employing explicit task-conditioning by the task-adaptive projection. TapNet consists of an embedding network $f_\theta$, the per-class reference vector set $\{\phi_n\}_1^N$, and the task-adaptive projection space $\mathbf{M}$. The embedder $f_\theta$ and the reference vectors $\{\phi_n\}_1^N$ are meta-trained across varying tasks, and not updated during evaluation. Otherwise, the task-adaptive projection space $\mathbf{M}$ which works as the classification space, is computed anew for every episode as a form of task-conditioning. With some arbitrary initial labeling on $\{\phi_n\}_1^N$, relabeling is not necessary throughout episodic training, as linear projection solutions always exist irrespective of particular reference vector labeling. When a task constructed by support set $\mathcal{S}$ and query set $\mathcal{Q}$ is given, the projection space $\mathbf{M}$ is computed based on the support set as follows. Using the embedding network $f_\theta$ and support set $\mathcal{S}$, the per-class network output averages $\mathbf{c}_n$ are obtained from the labeled samples in support set $\mathcal{S}$. Then the projection space is constructed to align $\mathbf{c}_n$ with the matching reference vector $\phi_n$ while distancing or disaligning it from all other non-matching references $\phi_{l \neq n}$. To achieve this, a modified reference vector is first formed as $\tilde{\phi}_n = \phi_n - \frac{1}{N-1}\sum_{l \neq n}\phi_l$. As the inner product between $\mathbf{c}_n$ and $\tilde{\phi}_n$ is effectively maximized in the process, the negative sign in front of the non-matching reference vectors would tend to disalign them from $\mathbf{c}_n$. The actual alignment is done by nulling every error vector $\boldsymbol{\epsilon}_n$ between the power-normalized $\tilde{\phi}_n$ and $\mathbf{c}_n$, i.e., $\boldsymbol{\epsilon}_n = \tilde{\phi}_n/\|\tilde{\phi}_n\| - \mathbf{c}_n/\|\mathbf{c}_n\|$. Nulling of the errors is essentially finding a projection matrix $\mathbf{M}$ such that $\boldsymbol{\epsilon}_n \mathbf{M} = \mathbf{0}$ for all $n$. Barring the trivial case of $\mathbf{M} = \mathbf{0}$, the resulting projection space $\mathbf{M}$ is a matrix whose columns span the task-adaptive classification space. The final classification is done by measuring the distance between the projected queries and the projected reference vectors.

### 2.3    TASK-ADAPTIVE CLUSTERING

Our *task-adaptive clustering* is also based on the embedding network $f_\theta$ and the per-class reference vector set $\{\phi_n\}_1^N$, both of which are meta-trained across episodes and *fixed* during evaluation. With the base learnable parts, task-conditioning of clustering space is done via linear projection to the null space of classification errors. For each episode, the projection space is initialized using only the labeled support samples as done in the work of TapNet, and it hosts task-adaptive clustering. Also, the clustering space is *iteratively* improved utilizing unlabeled samples. $\mathbf{M}$ will simply be referred to as projection space, and $\mathbf{M}(\mathbf{z})$ will represent projection of vector $\mathbf{z}$ in $\mathbf{M}$.

Now the TAC procedures based on repetitive reconstruction of the clustering space using unlabeled samples are described as follows: First, project the embedded features of the unlabeled samples into the new space $\mathbf{M}$ found above and estimate the soft labels (label probabilities) of the unlabeled

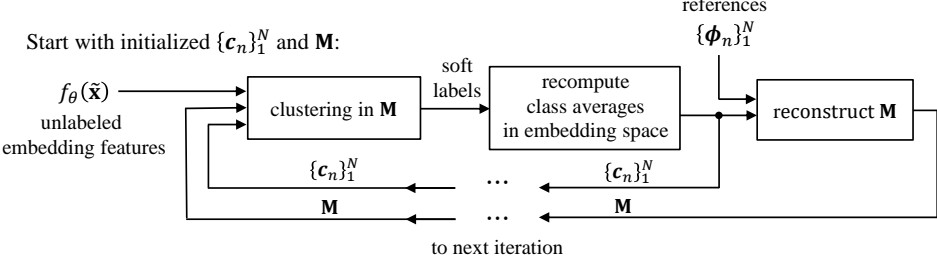

Figure 1: Iterative process for task-adaptive clustering

samples by clustering them around the projected per-class averages (see Figure 1). Recompute class averages in the original embedded space using the soft labels just estimated. Using the recomputed average with the references, reconstruct $\mathbf{M}$. In this sense, the projection space can be seen to be refined through an iterative process. The number of iterations in constructing projection space may be different between meta-training time and final testing time. More iterations would generally mean more aggressive adaptation to a given task. While a strong task-adaptation is desirable at deployment, it may result in excessive task-conditioning during meta-training, possibly hampering a proper buildup of inductive-bias.

Soft labels for unlabeled samples are estimated by the probability that an unlabeled sample $\tilde{\mathbf{x}}_j$ belongs to each class $n$: $p_{j,n} = \exp\left\{-d(\mathbf{M}\left(f_\theta(\tilde{\mathbf{x}}_j)\right), \mathbf{M}(\mathbf{c}_n))\right\}/\sum_l \exp\left\{-d(\mathbf{M}(f_\theta(\tilde{\mathbf{x}}_j)), \mathbf{M}(\mathbf{c}_l))\right\}$. With these probabilities, the centroids is refined by soft assignments of the embedded unlabeled samples, as done in (Ren et al., 2018):

$$\mathbf{c}_n^{\text{new}} = \frac{K\mathbf{c}_n + \sum_j p_{j,n} f_\theta(\tilde{\mathbf{x}}_j)}{K + \sum_j p_{j,n}}. \tag{1}$$

Finally, map the embedded features of the query samples and the reference vectors into the projection space, and adjust the parameters of the embedder and reference vectors using softmax based on Euclidean distances between projected query and reference vector pairs. Repeat this process until all episodes are exhausted.

The final few-shot classification procedure follows the same steps, up to the point where the mapped query is classified based on its distances to references in the final projection space. During model update, the Euclidean distances $d(\cdot, \cdot)$ between the projected query samples and references are used to compute the softmax function

$$\frac{\exp\left\{-d(\mathbf{M}(f_\theta(\hat{\mathbf{x}}_n)), \mathbf{M}(\boldsymbol{\phi}_n))\right\}}{\sum_l \exp\left\{-d(\mathbf{M}(f_\theta(\hat{\mathbf{x}}_n)), \mathbf{M}(\boldsymbol{\phi}_l))\right\}}. \tag{2}$$

The corresponding cross entropy loss function, averaged over all queries and classes, is used to update the embedder $f_\theta$ and references $\{\boldsymbol{\phi}_n\}_1^N$.

## 2.4 Handling Distractors with Task-Adaptive Clustering

We also need to handle unlabeled distractor samples coming from non-candidate classes. To this end, we make use of an additional centroid and a reference vector to represent the distractor samples. The approach is similar to that taken in (Ren et al., 2018), but here the additional pair of centroid and reference has a direct impact on the way projection space is constructed.

The above TAC procedure is modified as follows in handling distractor samples. We create an additional reference vector $\boldsymbol{\phi}_{N+1}$. Also, an initial centroid $\mathbf{c}_{N+1}$ for distractor samples is obtained by averaging all unlabeled samples in a given episode. When constructing a projection space $\mathbf{M}$, we utilize the additional error vector between $\mathbf{c}_{N+1}$ and $\boldsymbol{\phi}_{N+1}$. We also compute the probability $p_{j,N+1}$ that an unlabeled sample $\tilde{\mathbf{x}}_j$ is a distractor. For recomputation of class averages, a new centroid for distractors $\mathbf{c}_{N+1}^{\text{new}}$ is obtained by only considering soft labels $p_{j,N+1}$:

$$\mathbf{c}_{N+1}^{\text{new}} = \frac{\sum_j p_{j,N+1} f_\theta(\tilde{\mathbf{x}}_j)}{\sum_j p_{j,N+1}}. \tag{3}$$

At final classification time, projection space is recomputed with only $N$ original pairs of centroids and references, i.e., error vector between $\mathbf{c}_{N+1}^{\text{new}}$ and $\phi_{N+1}$ is not considered.

## 3 RELATED WORK

Semi-supervised learning is the learning method utilizing unlabeled examples in addition to labeled samples, and there exist many known approaches (Zhu, 2005; Chapelle et al., 2010). Among the various known semi-supervised learning methods, the self-training scheme of (Yarowsky, 1995; Rosenberg et al., 2005) is closely related to our method. In the self-training approach, the model is initially trained with labeled samples only, and prediction on unlabeled samples is made with this model. The prediction results are then utilized as labels of the unlabeled samples, and used for additional training of the model. The self-training approach outperforms the other semi-supervised learning methods when labeled data is scarce (Yarowsky, 1995; Triguero et al., 2015; Oliver et al., 2018), and also works well for training deep neural networks by using soft pseudo-labels for unlabeled samples (Lee, 2013). Our proposed task-adaptive clustering utilizes the prediction results for the unlabeled samples as soft labels in refining classification space, and this process also can be viewed as self-training with soft labels.

Semi-supervised few-shot learning was first proposed in (Ren et al., 2018). In that work, the soft $k$-means algorithm based on Prototypical Networks was suggested. In the soft $k$-means algorithm, the prototype for each class is generated with the labeled few shots of images first. Label predictions for the unlabeled samples are made using the prototypes, and in turn the results are used as soft labels for updating the class prototypes. Transductive Propagation Networks (TPNs) of (Liu et al., 2019) is a few-shot learner utilizing graph construction to propagate label information. TPN also exhibits a semi-supervised few-shot capability by propagating label information from labeled samples to unlabeled samples, and then to query samples. Our TAC pursues semi-supervised few-shot learning by updating class averages using the predicted soft labels for unlabeled samples, similar to Prototypical Networks based soft $k$-means. However, in TAC, label prediction is done in the projected space, not in the embedding space itself. We observe that unlabeled samples are better separated in the projected classification space than the embedding space, resulting in improved label prediction. Moreover, since the classification space itself can also be refined using the refined class averages, label prediction can be enhanced through the iterative process of constructing classification space. In the recent work of (Sun et al., 2019), semi-supervised few-shot learning with self-training is proposed. In that work, for each task, unlabeled samples are utilized to fine-tune the parameters of the model. For this purpose, confident unlabeled samples are first picked and a meta-trained soft-weighting-network computes the soft weights of the selected unlabeled samples. The computed weights are used to upweight beneficial examples and depress the less beneficial samples. This approach is closely related to the few-shot learning methods with task-specific fine-tuning (Finn et al., 2017; Ravi & Larochelle, 2017). Rather than adjusting model parameters, our TAC methods utilize unlabeled samples to update the alternative projection space.

TapNets of (Yoon et al., 2019) is a metric-based few-shot learning algorithm employing explicit task-conditioning via construction of task-adaptive projection space. In TapNets, the projection space is constructed using the support set samples and the reference vectors, and classification of query samples is done there. The present task-adaptive clustering scheme also relies on an alternative classification space obtained from the support set and reference vectors, but the clustering of unlabeled samples (and subsequent soft label estimation) and final classification of query samples are carried out in different spaces. The TAC space is first generated only with the support set and the references as done in TapNets, but the space itself gets improved iteratively by updating the class averages utilizing unlabeled samples. Moreover, in the TAC algorithm with a special handling of distractor samples from irrelevant classes, an additional cluster and reference pair is introduced to represent the distractor samples. The clustering space is generated using the additional cluster and reference together with the class averages and existing references, while the classification space is constructed without using the additional cluster or reference. Overall, TAC uses TapNets as base architecture, but a unique way of controlling the degree of task-conditioning between the meta-training phase and the evaluation phase provides a substantial gain beyond the TapNet baseline.

## 4    EXPERIMENT RESULTS

We evaluated the proposed semi-supervised few-shot learning algorithms with two benchmark datasets widely used to evaluate few-shot learning algorithms: 1) ***mini*ImageNet** suggested in (Vinyals et al., 2016) with the split introduced in (Ravi & Larochelle, 2017) and 2) ***tiered*ImageNet** suggested in (Ren et al., 2018), with the classes in test set highly distinct from those in training set.

### 4.1    EPISODE COMPOSITION FOR SEMI-SUPERVISED FEW-SHOT LEARNING

In our semi-supervised few-shot learning experiment, an additional dataset split is applied for each dataset as done in (Ren et al., 2018). We only use a small portion of label information from the dataset by dividing the samples in each class into disjoint labeled set and unlabeled set. For *tiered*ImageNet, 10% of samples per class constitute the labeled set while 90% of samples remain unlabeled. Meanwhile, for *mini*ImageNet, 40% of images are used as a labeled set, and remaining 60% of samples are used as an unlabeled set. The ratio of labeled samples for each dataset is based on the ratio used in (Ren et al., 2018).

For a given dataset and a split between labeled and unlabeled sets, episodes are created as follows. For meta-training, generation of episodes starts with sampling $N$ target classes from the meta-training set. Then, $K$ support set samples are selected from the labeled split of each target class. On the other hand, $u$ unlabeled images are chosen randomly from the unlabeled split of each class and form the unlabeled set. $q$ query samples per class are selected from the labeled split of each target class, disjointly with the support set. When the distractor samples are considered, $N_d$ different classes are additionally sampled from the meta-training set, and $d$ unlabeled images are sampled from the unlabeled split of each class. Consequently, the unlabeled set consists of $N \times u$ samples from $N$ candidate classes for classification and $N_d \times d$ images from the $N_d$ distractor classes. The test episodes are constructed in the same way as the training episodes but using the meta-test set. Note that both the number of classes $N$ and the number of distractor classes $N_d$ may be different between test episodes and training episodes.

### 4.2    EXPERIMENTAL SETTINGS

For the *mini*ImageNet experiment, we use an embedding network named CONV4, which is widely used in prior work of (Vinyals et al., 2016; Snell et al., 2017). For *tiered*ImageNet experiments, however, we added a $2 \times 2$ average pooling layer on top of this embedding network.

We evaluated the proposed method with semi-supervised 1-shot and 5-shot accuracies. For both 1-shot and 5-shot, we evaluate the algorithm with and without distractor samples. For the unlabeled set, we use 5 unlabeled samples per class for training and 20 unlabeled samples per class in evaluation. When the distractor samples are considered, the number of distractor class $N_d$ is the same as the number of target classes $N$ in an episode and the number of distractor samples per class $d$ is the same as the number of unlabeled samples per class $u$, for both training and evaluation.

Iterative task-adaptive clustering is employed in TAC experiments. We use just 1 iteration (single clustering) in meta-training and more iterations in evaluation generally . We believe that repetitive projection and clustering in episodic training sometimes result in excessive task-conditioning, which could hinder effective meta-learning of the model across tasks (i.e., effective buildup of inductive bias). The number of iterations in evaluation is chosen based on validation accuracy.

### 4.3    RESULTS

In Table 1, semi-supervised few-shot classification accuracies on *mini*ImageNet are presented. For the evaluation, we use classification accuracy averaged across 10 random splits between labeled and unlabeled samples. Accuracy of each split is computed with $3.0 \times 10^3$ episodes with 15 queries per class. For experiments without distractor, we can observe that our task-adaptive clustering achieves the best accuracy for 5-shot. In 1-shot measurements, our TAC shows the best accuracy while TAC distractor-aware projection (TACdap) shows an accuracy level slightly below that of TAC. With the distractor classes considered (1 or 5-shot w/D), our TACdap shows the best accuracy among the known methods. In Table 2, experimental results on *tiered*ImageNet are displayed. Again, without distractor samples, task-adaptive clustering yields the best results. For experiments with distractor

samples, TAC achieves the best 1-shot accuracy while TACdap shows the highest accuracy in the 5-shot case [1]. Notice that TAC or TACdap generally provides substantial gains over the TapNet baseline. For both PN and TapNet, baseline results are obtained using only the labeled samples. TapNet + Semi-Supervised Inference is a baseline which is meta-learned by supervised learning as TapNet and evaluated by task adaptive clustering utilizing the unlabeled samples. While this inference-only baseline shows the same level of accuracy with TAC in 5-shot *mini*ImageNet classification, generally our TAC or TACdap shows meaningful gains.

Table 1: Semi-supervised few-shot classification accuracies for 5-way *mini*ImageNet

| Methods | 1-shot | 5-shot | 1-shot w/D | 5-shot w/D |
|---|---|---|---|---|
| **PN baseline** (Ren et al., 2018) | 43.61% | 59.08% | 43.61% | 59.08% |
| **Soft *k*-Means** (Ren et al., 2018) | 50.09% | 64.59% | 48.70% | 63.55% |
| **Soft *k*-Means + Cluster** (Ren et al., 2018) | 49.03% | 63.08% | 48.86% | 61.27% |
| **Masked Soft *k*-Means** (Ren et al., 2018) | 50.41% | 64.39% | 49.04% | 62.96% |
| **TPN** (Liu et al., 2019) | 52.78% | 66.42% | 50.43% | 64.95% |
| **MetaGAN + RN** (Zhang et al., 2018) | 50.35% | 64.43% | - | - |
| **IMP**[†] (Allen et al., 2019) | - | - | 49.2% | 64.7% |
| **TapNet baseline** | 49.58% | 66.86% | 49.58% | 66.86% |
| **TapNet + Semi-Supervised Inference** | 53.19% | **69.53**% | 50.53% | 66.93% |
| **TAC** (Ours) | **55.50**% | **69.21**% | 50.95% | 66.39% |
| **TACdap** (Ours) | 52.60% | 69.05% | **51.56**% | **67.75**% |

*Due to space limitation, 95% confidence intervals of the reported accuracies are given in Appendix.
† The result from (Allen et al., 2019) is based on the setting with 5 unlabeled samples in evaluation.

Table 2: Semi-supervised few-shot classification accuracies for 5-way *tiered*ImageNet

| Methods | 1-shot | 5-shot | 1-shot w/D | 5-shot w/D |
|---|---|---|---|---|
| **PN baseline** (Ren et al., 2018) | 46.52% | 66.15% | 46.52% | 66.15% |
| **Soft *k*-Means** (Ren et al., 2018) | 51.52% | 70.25% | 49.88% | 68.32% |
| **Soft *k*-Means + Cluster** (Ren et al., 2018) | 51.85% | 69.42% | 51.36% | 67.56% |
| **Masked Soft *k*-Means** (Ren et al., 2018) | 52.39% | 69.88% | 51.38% | 69.08% |
| **TPN** (Liu et al., 2019) | 55.74% | 71.01% | 53.45% | 69.93% |
| **TapNet baseline** | 51.84% | 69.14% | 51.84% | 69.14% |
| **TapNet + Semi-Supervised Inference** | 55.66% | 71.54% | 51.84% | 69.05% |
| **TAC** (Ours) | **58.46**% | **72.05**% | **54.80**% | 69.35% |
| **TACdap** (Ours) | 56.06% | 71.37% | 53.67% | **70.72**% |

*95% confidence intervals are again given in Appendix.

## 4.4 ANALYSIS OF PROJECTION SPACE

**tSNE plot.** For 5-way, 1-shot *tiered*ImageNet experiments without distractor, Figure 2 illustrates the embedding space and the iteratively reconstructed TAC space with t-SNE. For better illustration, we visualize each space with 80 unlabeled samples per class. In the t-SNE plot, we display the unlabeled samples correctly clustered to each class with colored dots, while the incorrectly clustered samples are marked as gray dots. The number marked beside each cluster indicates the number of correctly-clustered unlabeled samples for each class. We can see that the unlabeled samples result in better cluster separation in the TAC space than the embedding space, and the number of incorrectly clustered samples decreases as the TAC space is updated repeatedly.

**Performance improvement via iterative task-adaptive clustering.** In Figure 3, classification accuracies are plotted versus the number of iterations. For 5-way 1-shot *mini*ImageNet classification,

---

[1]We do not include evaluation results of (Sun et al., 2019) because their base embedder is pretrained on fully-labeled training sets of *mini/tiered*ImageNet datasets; we feel that direct comparison would not be fair at this point.

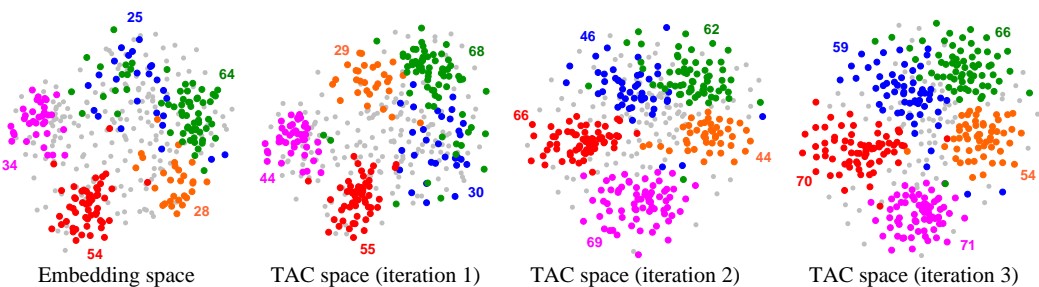

| Embedding space | TAC space (iteration 1) | TAC space (iteration 2) | TAC space (iteration 3) |

Figure 2: tSNE visualization of iterative improvements of clustering space

we compare the proposed TAC with Soft $k$-Means of (Ren et al., 2018). We can see that the accuracy of TAC steadily increases as the number of iterations grows, before saturating; on the other hand, the accuracy of Soft $k$-Means of (Ren et al., 2018) fluctuates with the number of iterations. For the *tiered*ImageNet classification, comparison is made against Masked Soft $k$-Means instead [among methods of (Ren et al., 2018), Soft $k$-Means and Masked Soft $k$-Means show the best performance for *mini*ImageNet and *tiered*ImageNet classification, respectively]. Similarly, there is no gain from the iteration for Masked Soft $k$-Means, while significant gain is seen with the iteration for TAC. Note that more iteration here provides a deeper level of task-conditioning. The results of Figure 3 suggest that TAC iterations up to 4 would be beneficial.

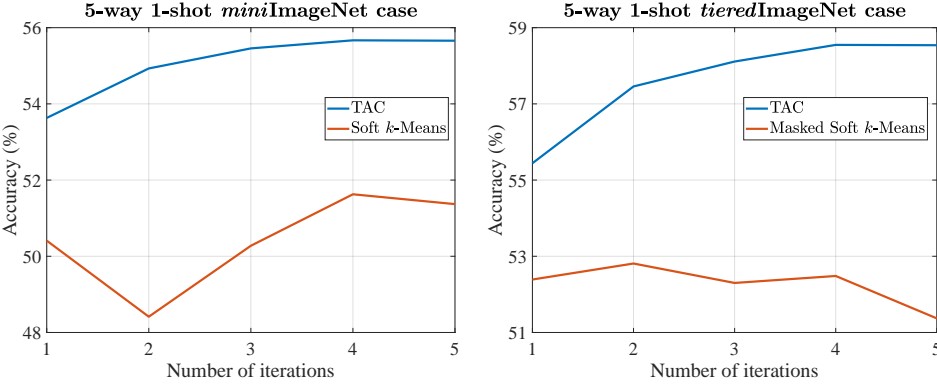

Figure 3: Semi-supervised few-shot classification accuracies versus number of iterations

## 4.5 SEMI-SUPERVISED FEW-SHOT LEARNING WITH REALISTIC UNLABELED SET

Experimental settings and episode composition for semi-supervised few-shot learning were first suggested in (Ren et al., 2018). However, we found that the episode composition of prior work may not be realistic. In (Ren et al., 2018), the unlabeled set of each episode consists of $N \times u$ target class unlabeled samples and $N_d \times d$ distractor samples, where $N = N_d$ and $u = d$. In this setting, the unlabeled set contains exactly the same number of unlabeled samples for each class. Also, when distractor samples are considered, they are selected only from the $N_d$ classes, and the unlabeled set has the same number of target class samples and distractor class samples. To construct the unlabeled set in this structured manner, we would need label information of the unlabeled split of the dataset, which is not realistic. Here we propose more realistic episode composition for semi-supervised few-shot learning. We construct the unlabeled set in an uneven manner, with some randomness relative to the evenly composed unlabeled set. We select candidate-class unlabeled samples randomly but non-uniformly from the unlabeled splits of $N$ candidate classes. As for distractor samples, we sample them randomly from the unlabeled split of whole classes except the candidate classes. We do not fix the number of samples per each class, nor the number of distractor classes. We only determine the total number of candidate-class unlabeled samples and distractor samples. Furthermore, it is more natural for an unlabeled set to include more distractor samples than candidate-class samples.

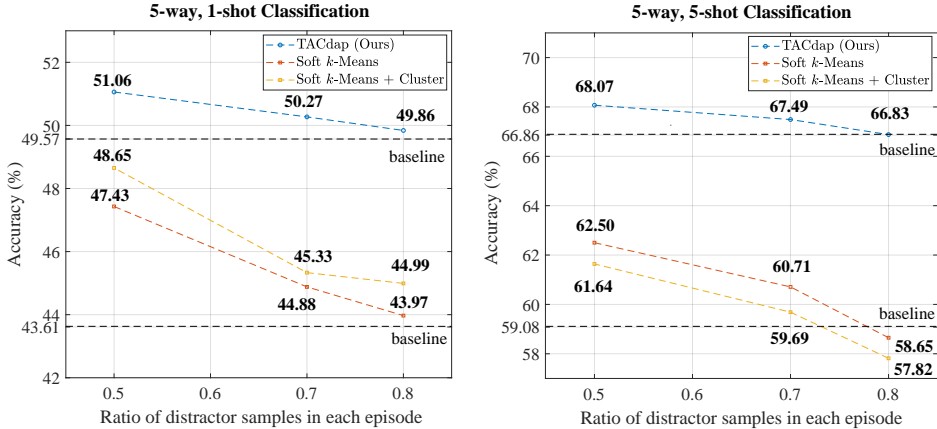

Figure 4: Semi-supervised few-shot classification accuracies of the proposed TAC distractor-aware projection and with a growing fraction of distractor samples in each episode

We perform experiments with the episodes constructed by this more realistic unlabeled set composition. In the experiment, we use 50 unlabeled samples per episode in training, and 200 unlabeled samples per episode in evaluation. It is the same as the total number of unlabeled samples in the experiment in (Ren et al., 2018). We compare our TAC distractor-aware projection with the Soft $k$-Means and Soft $k$-Means + Cluster semi-supervised few-shot learner of (Ren et al., 2018). The 1-shot and 5-shot accuracies with varying ratios of distractor samples are shown in Figure 4. The accuracy degrades as the ratio of distractor samples increases. We can observe that the 1-shot and 5-shot accuracies of Soft $k$-Means degrade from 48.70% and 63.55% to 47.43% and 62.50% when unevenly composed episodes are applied. Also, we can observe that the 5-shot accuracy of Soft $k$-Means and Soft $k$-Means + Cluster drops below the supervised baseline when 80% of unlabeled samples are distractor samples while our TACdap shows the same level of accuracy as the supervised baseline.

## 5 CONCLUSION

We proposed a semi-supervised few-shot learning method utilizing task-adaptive clustering, which performs clustering in a new projection space for providing a controlled amount of task-conditioning. The TAC space is first constructed to align the class averages with the meta-trained reference vectors, and then gets iteratively refined with the aid of the unlabeled samples clustered in the TAC space. The TAC-based algorithms shows state-of-the-art semi-supervised few-shot classification accuracies on the *mini*ImageNet and *tiered*ImageNet datasets, with and without distractor samples. The proposed TAC algorithms also show graceful degradation with a growing portion of distractor samples in the unlabeled sample set.

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

## A  HYPERPARAMETER SETTINGS

Tables 3 and 4 show the hyperparameter settings of our experiments in the main paper. We utilize Adam optimizer with an initial learning rate of $10^{-3}$ for all experiments. We cut the learning rate by a factor of 10 once after training $2.5 \times 10^4$ episodes or twice after training $2.5 \times 10^4$ and $2.75 \times 10^4$ episodes. For regularization, the $l2$ weight decay with an optimized decay rate is also applied in all experiments. In the meta-training for all cases, 10-way training is adopted. The number of queries per class is 12 for 1-shot training, and 8 for 5-shot training. Some 1-shot models adopt higher-shot training. For both *mini*ImageNet and tieredImageNet experiments, TAC 1-shot and TAC 1-shot w/D models adopt 5-shot training with 8 queries.

Table 3: Hyperparameter settings for *mini*ImageNet experiments

| Model | Training shot | $N$ | $N_q$ | lr decay step | $l2$ decay rate | $N_{\text{iter}}$ |
|---|---|---|---|---|---|---|
| **TAC** 1-shot | 5 | 10 | 8 | 25000 & 27500 | 3e-4 | 5 |
| **TACdap** 1-shot | 1 | 10 | 12 | 25000 & 27500 | 3e-4 | 3 |
| **TAC** 5-shot | 5 | 10 | 8 | 25000 & 27500 | 3e-4 | 2 |
| **TACdap** 5-shot | 5 | 10 | 8 | 25000 & 27500 | 3e-4 | 2 |
| **TAC** 1-shot w/D | 5 | 10 | 8 | 25000 | 3e-4 | 3 |
| **TACdap** 1-shot w/D | 1 | 10 | 12 | 25000 & 27500 | 3e-4 | 2 |
| **TAC** 5-shot w/D case | 5 | 10 | 8 | 25000 | 3e-4 | 1 |
| **TACdap** 5-shot w/D | 5 | 10 | 8 | 25000 & 27500 | 3e-4 | 1 |

Table 4: Hyperparameter settings for *tiered*ImageNet experiments

| Model | Training shot | $N$ | $N_q$ | lr decay step | $l2$ decay rate | $N_{\text{iter}}$ |
|---|---|---|---|---|---|---|
| **TAC** 1-shot | 5 | 10 | 8 | 25000 & 27500 | 3e-5 | 5 |
| **TACdap** 1-shot | 1 | 10 | 12 | 25000 & 27500 | 3e-5 | 4 |
| **TAC** 5-shot | 5 | 10 | 8 | 25000 & 27500 | 3e-5 | 3 |
| **TACdap** 5-shot | 5 | 10 | 8 | 25000 & 27500 | 3e-5 | 2 |
| **TAC** 1-shot w/D | 5 | 10 | 8 | 25000 & 27500 | 3e-5 | 2 |
| **TACdap** 1-shot w/D | 1 | 10 | 12 | 25000 & 27500 | 1e-5 | 3 |
| **TAC** 5-shot w/D | 5 | 10 | 8 | 25000 & 27500 | 3e-5 | 1 |
| **TACdap** 5-shot w/D | 5 | 10 | 8 | 25000 & 27500 | 3e-5 | 1 |

## B  CONFIDENCE INTERVALS

Table 5 and 6 show the classification accuracies of our method with 95% confidence intervals for *mini*ImageNet and *tiered*ImageNet datastets, respectively.

## C  ABLATION STUDY

### C.1  NUMBER OF CLUSTERING ITERATIONS IN EVALUATION

The number of iterations for clustering the unlabeled sample is optimized for each experiment. Figures 5 and 6 show *mini*ImageNet and *tiered*ImageNet classification accuracies on validation set with varying numbers of iterations. We fixed the iteration number to 1 during episodic meta-training. Iteration 1 means that clustering of unlabeled samples is done once in TAC space. We measured the classification accuracies with iteration numbers ranging from 1 to 5. For every case, we marked the number of iterations yielding the best accuracy with a solid dot. The iteration number used in the main paper is set to be the best choice observed in this validation process. Note that the aggressive iteration is advantageous for 1-shot cases (3 to 5 iterations result in considerable gains). On the other hand, the iterative process is less beneficial to 5-shot or distractor sample cases (w/D). Also, in general TAC seems to benefit somewhat more from iterations than TACdap.

Table 5: Semi-supervised few-shot classification accuracies for 5-way *mini*ImageNet

| Methods | 1-shot | 5-shot | 1-shot w/D | 5-shot w/D |
|---|---|---|---|---|
| **PN baseline** | 43.61±0.27% | 59.08±0.22% | 43.61±0.27% | 59.08±0.22% |
| **Soft *k*-Means** | 50.09±0.45% | 64.59±0.28% | 48.70±0.32% | 63.55±0.28% |
| **Soft *k*-Means + Cluster** | 49.03±0.24% | 63.08±0.18% | 48.86±0.32% | 61.27±0.24% |
| **Masked Soft *k*-Means** | 50.41±0.31% | 64.39±0.24% | 49.04±0.31% | 62.96±0.14% |
| **TPN** | 52.78±0.27% | 66.42±0.21% | 50.43±0.84% | 64.95±0.73% |
| **MetaGAN + RN** | 50.35±0.23% | 64.43±0.27% | - | - |
| **IMP** | - | - | 49.2±0.7% | 64.7±0.7% |
| **TapNet baseline** | 49.58±0.20% | 66.86±0.16% | 49.58±0.20% | 66.86±0.16% |
| **TAC** (Ours) | **55.50±0.27**% | **69.21±0.19**% | 50.95±0.23% | 66.39±0.19% |
| **TACdap** (Ours) | 52.60±0.25% | 69.05±0.14% | **51.56±0.17**% | **67.75±0.26**% |

Table 6: Semi-supervised few-shot classification accuracies for 5-way *tiered*ImageNet

| Methods | 1-shot | 5-shot | 1-shot w/D | 5-shot w/D |
|---|---|---|---|---|
| **PN baseline** | 46.52±0.52% | 66.15±0.22% | 46.52±0.52% | 66.15±0.22% |
| **Soft *k*-Means** | 51.52±0.36% | 70.25±0.31% | 49.88±0.52% | 68.32±0.22% |
| **Soft *k*-Means + Cluster** | 51.85±0.25% | 69.42±0.17% | 51.36±0.31% | 67.56±0.10% |
| **Masked Soft *k*-Means** | 52.39±0.44% | 69.88±0.20% | 51.38±0.38% | 69.08±0.25% |
| **TPN** | 55.74±0.29% | 71.01±0.23% | 53.45±0.93% | 69.93±0.80% |
| **TapNet baseline** | 51.84±0.16% | 69.14±0.13% | 51.84±0.16% | 69.14±0.13% |
| **TAC** (Ours) | **58.46±0.19**% | **72.05±0.15**% | **54.80±0.20**% | 69.35±0.17% |
| **TACdap** (Ours) | 56.06±0.18% | 71.37±0.12% | 53.67±0.18% | **70.72±0.13**% |

## C.2 NUMBER OF UNLABELED SAMPLES IN EVALUATION

Figure 7 shows *mini*ImageNet classification accuracies of the proposed methods with an increasing number of unlabeled samples in evaluation. All models are trained and tested with single iteration, and trained with 5 unlabeled samples per class. We can see that the test accuracy gradually increases as the number of unlabeled samples grows. Note that when the number of unlabeled samples is low, the accuracy of TACdap is always higher than the accuracy of TAC. Moreover, for 1-shot cases, the accuracies of TACdap with 1 unlabeled sample is fairly close to the accuracies with many unlabeled samples.

## C.3 CLASSIFICATION ACCURACY AND CROSS ENTROPY VERSUS NUMBER OF ITERATIONS.

In Figure 8, classification accuracies and cross entropy are plotted versus the number of iterations for 5-way, 1-shot cases of *mini*ImageNet and *tiered*ImageNet experiments without distractor. We consider two cases for clustering at evaluation time: in projection space and in embedding space. The final classification is done in the task-adaptive projection space. Cross entropy $L_u$ is obtained from the label probabilities. As the number of iterations grows, classification accuracy improves and cross entropy decreases for both cases, i.e., the semi-supervised classification performance is enhanced and the soft labels become more reliable. The initial gap between the two cases actually comes from the clustering in projected space rather than embedding space. Also, as iteration grows from 1 to 5, the gap is actually seen to gradually increase: from 0.90% to 1.19% and from 1.27% to 1.65% for *mini*ImageNet and *tiered*ImageNet, respectively. The results clearly show that clustering (and subsequent soft-labeling) is much better off done in the projection space than in the original embedding space for leveraging unlabeled samples.

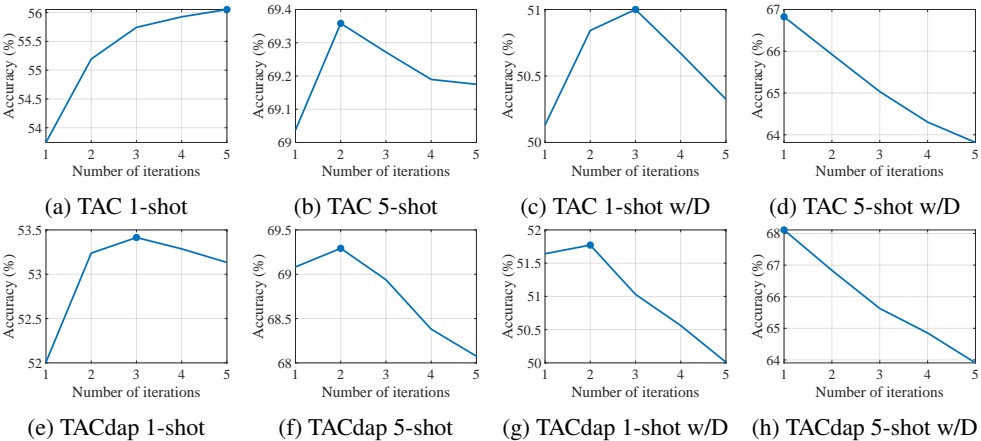

Figure 5: *mini*ImageNet classification accuracies vs iteration number

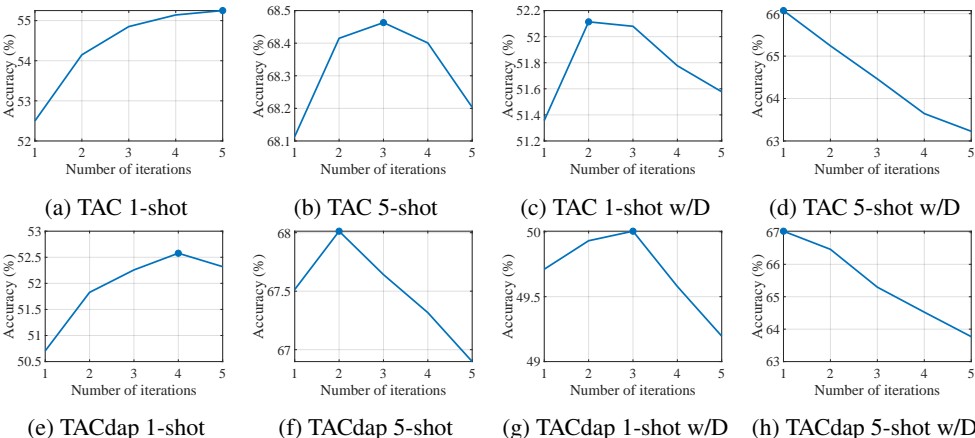

Figure 6: *tiered*ImageNet classification accuracies vs iteration number

### C.4 CHOICE OF CLUSTERING SPACE IN META-TRAINING AND IN EVALUATION

In the main paper, we showed that clustering in the TAC space results in better classification accuracies than clustering in the embedding space. A question may arise: Is clustering in TAC space still beneficial even when clustering is rather done in the embedding space during meta-training? In other words, how would the learner fare if the episode training condition deviates from the actual few-shot evaluation condition in terms of clustering space. Note that the principle established with regards to episodic training is that the training condition must match the evaluation condition. Interestingly, for the TAC methods, clustering in the embedding space during meta-training did not affect final performance even as evaluation is based on clustering in TAC space. See the comparison results in Figure 9. Clustering in the embedding space in fact gave similar (or slightly better at low iteration numbers) results compared to clustering in projection space, when it comes to meta-training. As for better clustering at evaluation time, TAC space is the better choice than original embedding space.

### C.5 NUMBER OF CLUSTERING ITERATIONS IN META-TRAINING

The effect of varying the iteration number in meta-training is also investigated. We wish to understand whether task-adaptation through iterative projection and clustering in meta-training is productive or not. It may be possible that strong adaptation for every episode during meta-training may hinder the process of building a proper inductive bias in the model. For the results obtained in the main manuscript, we adopted 1 iteration of clustering in TAC space. Here we present measured

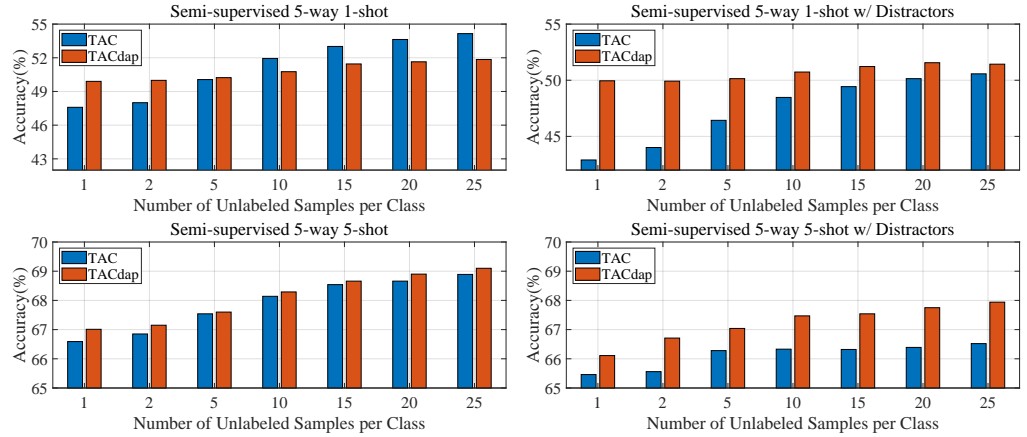

Figure 7: *mini*ImageNet classification accuracy versus number of unlabeled samples

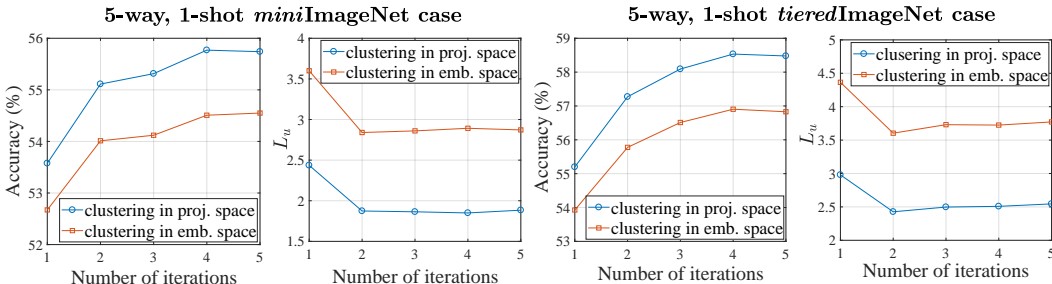

Figure 8: Classification accuracy and cross entropy vs number of iterations

*tiered*ImageNet classification accuracies for TAC and TACdap methods with a varying number of iterations in meta-training.

See Figure 10. For TAC methods, a more aggressive adaptation through an increasing number of iterations during meta-training does not result in significant gains. Note that 0 iteration here means that clustering is done in the embedding space.

For TACdap, on the other hand, as the number of iterations grows, performance is substantially degraded. For 1-shot experiments, accuracy is consistently degraded as the number of iterations grows to more than 1. Also for 5-shot experiments, when we adopt iterative clustering in TAC space, classification accuracy decreases steadily with iteration. Thus, for TACdap, multiple rounds of clustering during meta-training are actually harmful.

Overall, during meta-training, iterative projection followed by subsequent clustering is not beneficial for our TAC-based methods. We believe that multiple rounds of projection/clustering at train time may cause excessive task-conditioning, which in turn would prevent the machine from developing a healthy level of inductive bias for meta-learning.

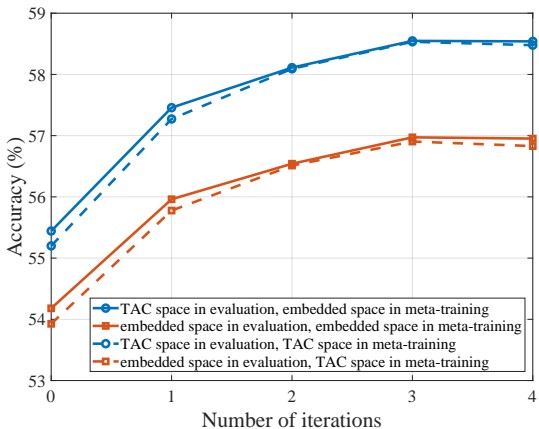

Figure 9: Classification accuracy on *tiered*ImageNet vs space where clusters form: TAC space versus embedded space, in meta-training and in evaluation

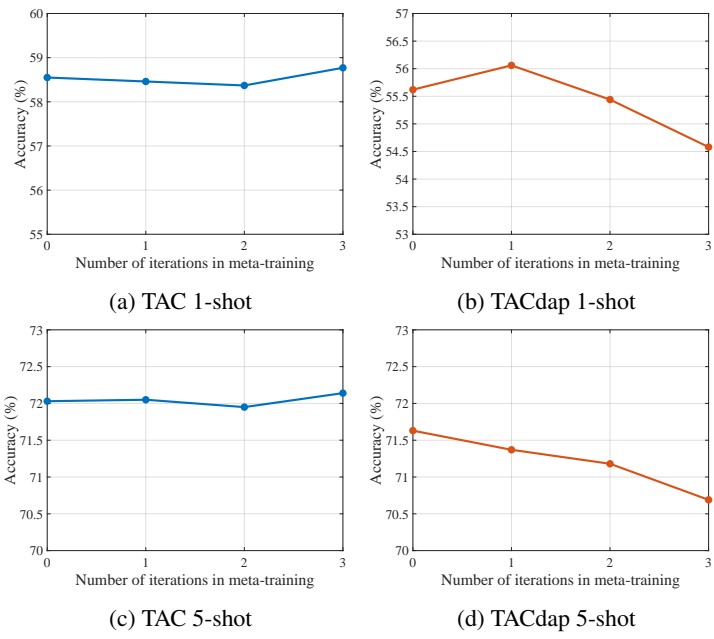

Figure 10: *tiered*ImageNet classification accuracy versus number of iterations in meta-training

