# OpenReview forum: "Semi-Supervised Few-Shot Learning with a Controlled Degree of Task-Adaptive Conditioning"
_ICLR.cc/2020/Conference — Reject_

### Official Review · AnonReviewer2 · 2019-10-22
**Official Blind Review #2**

**Rating:** 3

**Review:**

This paper presents a semi-supervised few-shot learning method that combines the TapNet architecture with the soft-clustering approach in the original semi-supervised few-shot learning method of Ren et al. 2018. Results are reported on mini-ImageNet and tiered-ImageNet, demonstrating superior results to Ren et al. 2018 and some more recent work.

Though the results are strong, I'm personally leaning towards rejecting this submission. The main reason is that the contribution is an arguably simple combination of 2 methods (TapNet + Ren et al. 2018). Specifically, I notice that the difference between the performance of the TapNet and Prototypical Networks baselines (6%, 7.5% for 1 and 5 shot on miniImageNet, and 5% and 3% for 1 and 5 shot on tieredImageNet) is roughly also found in the difference between TAC and Soft- k-Means (Ren et al. 2018). This suggests that most of the benefits found in TAC can simply be explained by the adoption of the TapNet architecture. And I consider this proposed extension to semi-supervised few-shot learning of TapNet to be pretty straightforward.

I don't really expect the authors can improve on the above point in a revision, so I doubt I'll be convinced to change my score. However, if the authors believe that I've missed a subtlety of their method that is less technically straightforward than my analysis suggests, of course I'd like to know.

**Experience Assessment:**

I have published in this field for several years.

**Review Assessment: Checking Correctness Of Derivations And Theory:**

N/A

**Review Assessment: Checking Correctness Of Experiments:**

I carefully checked the experiments.

**Review Assessment: Thoroughness In Paper Reading:**

I read the paper thoroughly.

---

> ### Author Response · Authors · 2019-11-13
> **Response to Official Blind Review #2**
>
> We agree that the proposed method is indeed a combination of TapNet and soft-clustering. However, we maintain that the proposed method is not an easily anticipated extension/combination of prior work. Note that the proposed iterative updates of projection space provide an increasing degree of task-conditioning, a newly introduced notion, which we believe is very significant. Ren et al. also attempt iterative updates of class prototypes, but there is no evidence of performance improvement with this process. In fact, our own evaluation indicates that the Ren et al. method does not provide performance gain with iterative clustering (see Figure 3 in our paper). Note that in Ren et al. (and also in the original Prototypical Networks) there is no explicit mechanism for task-specific conditioning, whereas in our method the projection space reflects task-conditioning (as it changes with a new set of tasks). In this sense, an iterative strengthening of projection space, as introduced in this paper, means a stronger task-conditioning.
>
> We also would like to clarify that the performance gap between TAC and Soft-k-Means is not simply due to the performance advantage of the baseline TapNet. When soft-clustering is simply combined with TapNet (clustering in embedding space and a subsequent classification in projection space), 1-shot and 5-shot accuracies for miniImageNet are 52.82% and 68.76%, respectively. These numbers, when TAC is used, become 55.50% and 69.21%. For tieredImageNet, these numbers are 54.18% and 71.84% for the simple combination versus 58.48% and 72.05%, respectively, for the proposed TAC algorithm. In both datasets, while 5-shot performances show small improvements, 1-shot accuracy gains are substantial. Given the importance of fewer-shot results, we feel that the significance of the proposed method is high.

---

### Official Review · AnonReviewer1 · 2019-10-23
**Official Blind Review #1**

**Rating:** 6

**Review:**

Summary
========
This paper tackles the more realistic variant of few-shot classification where a large portion of the available data is unlabeled, both at meta-training time in order to meta-learn a learner capable of fast adaptation, as well as meta-test time for adapting said learner to each individual held-out task.

Their approach is based on TapNet, a model which constructs a task-specific projection based on the support set of each episode, and then classifies each query according to its distance to each class prototype in the projected space. The projection is computed so that the class prototypes of the episode (averaged support examples) are well-aligned with a set of meta-learned references. Intuitively, those references learn to be far from each other, so that aligning the prototypes with them leads to a space where the episode’s classes are well separated, allowing for easier classification between them.

They then extend this model to incorporate unlabeled examples by performing task projection as follows: 1) the projection is computed so as to align the initial prototypes (computed only using labeled examples) to the meta-learned references. 2) In that projected space, each unlabeled example is assigned a predicted label based on its proximity to the projected prototypes. 3) Then, back at the original space, those predicted labels of the unlabeled examples are used to refine the class prototypes (weighted average as in Ren et al). 4) The projected space so that the *refined* prototypes are best aligned with the meta-learned references. 5) Possibly repeat 2-4 (at meta-test time).

Experimentally, they outperform recent approaches to semi-supervied few-shot classification on standard benchmarks, though not by far. Perhaps more interestingly, their performance degrades less than that of Ren et al as the distractor ratio increases, and they show that their method benefits from additional steps of task adaptation, whereas that of Ren et al reaches its performance limits after the first step of soft clustering.

High-level comments
==================
A) Ablation: An interesting ablation would be: instead of going back and forth between the embedding space and the projected space, the task adaptation happens only in the initially-computed projection space (i.e. the one computed based on the labeled data only). This would amount to: computing the projection space, and then performing a few steps of soft clustering, similar to Ren et al. in that space. This would help determine how beneficial it is to re-compute that projection space according to the current ‘best guess’ of where the class prototypes lie at each iteration. The way I see it, it is an empirical question whether the initially computed projection space already sufficiently separates the classes or not. I assume this would also lead to a more computationally efficient solution?

B) Handling distractors: In the case of distractors, they use an additional centroid (and a corresponding additional reference vector) for the purpose of ‘capturing’ the unlabeled examples that don’t belong to candidate classes. I find the initialization of this strange: this additional centroid is computed as the mean of all unlabeled examples, and the initial projection construction is influenced by a term that matches this centroid to a corresponding reference. This would mean, however, that even the unlabeled examples which do indeed belong to one of the candidate classes end up far from those classes in the projected space, in order to be close to the designated extra reference in that space. We know that this is not ideal, since we assume that some unlabeled examples do belong to the same classes as the labeled ones. Is there a way to quantify how severely this affects the quality of this initial projection? I would also be curious about the meta-learned location of the extra reference. Does it end up being roughly in the center of the references corresponding to the labeled classes?

C) Inference-only baselines. Ren et al. experimented with inference-only baselines: meta-learning happens only using the labeled subset, and the proposed clustering approach only takes place at meta-test time. In this case this would amount to meta-training a standard TapNet and then performing the proposed refinement only in test episodes. This is interesting as it allows to understand the importance of learning an embedding end-to-end for being more appropriate for unlabeled example refinement. It is not obvious that this is required, so I would be curious to see these results too. (This differs from the reported TapNet baseline in that at meta-test time it would make use of the proposed semi-supervised refinement).

Clarity / quality of presentation:
============================
D) A lot of emphasis is placed on the ability of the proposed method to control the degree of task conditioning. I would like to emphasize that this is not something that previous methods lack. Ren et al.’s approach could also perform multiple steps of clustering for example. Whether or not this is beneficial is an empirical question, but I wouldn’t say that the proposed method does something special to “control” how much adaptation is performed.

E) It would have been useful to have a separate section or subsection that explains TapNet, since this is the model that the proposed method is built upon. Instead, the authors blend the explanation of TapNet in the same section as their method which makes it hard to understand the approach and to separate the contribution of this method from TapNet.

G) The length of the paper is over the recommended pages, and I did feel that a few parts were too lengthy or repetitive (e.g. the last paragraph of the introduction described the model in detail. I felt that it would have been more appropriate to give the higher level intuition there and leave the detailed description for the next section).


**Experience Assessment:**

I have published one or two papers in this area.

**Review Assessment: Checking Correctness Of Derivations And Theory:**

N/A

**Review Assessment: Checking Correctness Of Experiments:**

I assessed the sensibility of the experiments.

**Review Assessment: Thoroughness In Paper Reading:**

I read the paper thoroughly.

---

> ### Author Response · Authors · 2019-11-13
> **Response to Official Blind Review #1 (1/2)**
>
> A) This is an important question, and in answering it we first note that in any TapNet-based semi-supervised method, the projection space must always be updated before final classification is done if the refined class centroids are to be utilized. This is because in TapNet final classification is done by comparing the query with the stand-alone per-class reference vectors on the projection space (not with the prototypes themselves). Since the reference vectors do not change during meta-testing, the only way the refined prototypes could influence classification accuracy is through projection space update. This means that to employ the proposed method, there must be at least one projection space update right before final classification. Now having said that, we conducted an ablation study on what if the iterative clustering is done through repetitive soft-clustering on the initial projection space, with only one-time projection space update in the end before final classification (which is done using the per-class reference vectors now projected onto this new space). The resulting miniImageNet 1-shot and 5-shot accuracies without distractor are 55.61% and 69.05%, which are similar to the reported results conducting projection update with each new clustering. For the tieredImageNet, the 1-shot and 5-shot accuracies without distractor are 58.89% and 71.99%, which are also similar to the accuracies of TAC with projection update for each new clustering. We note that even with the one-time projection update, the projection space undergoes a significant change reflecting the refinements in cluster centroids. Our short answer to the question on whether the initially computed projection space already sufficiently separates the classes or not is that the final projection space (whether obtained through a series of updates or computed at once) provides a considerably better separation than the initial projection space, as seen from the reported results of accuracy versus iteration number (blue curves in Figures 3 and 8). We also note that we already provided an ablation study on doing iterative soft clustering in embedded space and then find a projection space at the end where the classification is done. The results in Figure 8 (red curves) show that this approach is significantly worse than our proposal. In summary, the projection is better be done in the beginning and then at least once more at the end of the iterative clustering process.
>
>
> B) The reviewer is correct in that the additional centroid we introduce reflects even the samples belonging to valid classes. However, it is easy to see that said samples are actually closer to their correct centroids than this additional centroid, because the new centroid necessarily averages samples from all classes, including the distractors. In other words, an unlabeled sample belonging to some valid class will likely be viewed as a member of that cluster eventually, even though this sample influenced the computation of the additional centroid. In the work of Ren et al. the “zero prototype” is added for handling distractor samples. Even the arbitrarily chosen zero-vector seems to work, although ours is slightly better. For 1-shot and 5-shot miniImageNet with distractor samples, when we initialize the prototype for distractor as a zero vector, the accuracies are 51.35% and 67.10%, while the presented 1-shot and 5-shot accuracies are slightly better at 51.56% and 67.75%, respectively.
>
>
> C) The reviewer raises an interesting point. For miniImageNet, the suggested inference-only baseline accuracies are 53.19% for 1-shot and 69.53% for 5-shot, when the distractor is not considered. These compare to 55.50% for 1-shot and 69.21% for 5-shot in our reported results. Thus, for 1-shot, training consistently between meta-learning and meta-testing is somewhat better, while for 5-shot iterative clustering at meta-testing only gives a slightly better or similar result. On the other hand, for tieredImageNet, the inference-only baseline accuracies are 55.66% and 71.54% for 1-shot and 5-shot, when the distractor is not considered. These accuracies improve to 58.46% and 72.05% in our results. Thus, for the tieredImageNet case, training consistently between meta-learning and meta-testing seems to show definitely better results. With distractor samples, our TACdap consistently shows better results than the inference-only baselines on both miniImageNet and tieredImageNet experiments. We have included all these results for inference-only baselines in the uploaded version of paper (as TapNet + Semi-Supervised Inference in Tables 1 and 2).

---

> ### Author Response · Authors · 2019-11-13
> **Response to Official Blind Review #`1 (2/2)**
>
> D) We understand that Ren et al. also talks about repetitive clustering. But we wanted to emphasize that in their work multiple steps of clustering did not help (as confirmed in Figure 3 of our paper) whereas in our case they did. To us, what this means is that our method allows different levels of task-conditioning through a repetitive update of projection space. Since the formation of the projection space represents task-conditioning, we felt that our approach introduced a way to control the amount of task-conditioning.
>
> E) We now added a separate section on TapNet baseline with a better explanation in the revised version. This hopefully makes the contribution of this paper clearer.
>
> G) We have shortened the introduction as suggested.

---

### Official Review · AnonReviewer3 · 2019-10-24
**Official Blind Review #3**

**Rating:** 1

**Review:**

Paper Summary:

This paper extends TapNet, a few short learning approach, to the setting of semi-supervised few shot learning. For that, it borrows the idea of soft-labeling from Ren et al 2018, a semi-supervised few shot learning algorithm. Experiments over mini and tiered imagenet compare the approach with alternatives.

Review Summary:

The paper could be clearer. The main idea of the paper and its goal/motivation is not concisely given in the intro and abstract. The paper refers to a lot of terminology from TapNet without defining them which makes it difficult to understand. I am not an expert in the field and I have a hard time judging whether the empirical improvement are exceptional but I would advocate against accepting the paper based on clarity alone at this point. Moreover the contribution -- adding soft labeling to TapNet -- seems modest.

Detailed Review:

The abstract should not be a sketch of the proposed algorithm. It should highlight: what is the problem, what is the key idea of what you propose to address it (one sentence), why is it original and better than prior work, what is the foreseen impact of this work.

The terminology specific to your problem and family of model need to be defined: what is the clustering space? which clustering? what is the classification space? what is a per class network? None of these concepts are defined.

In section 2.2, please rephrase "Nulling of the errors is essentially finding a projection matrix M such that epsilonM = 0 for all n. The resulting M is a matrix whose columns span the task-adaptive classification space" as "We identify M such that (i)  epsilonM = 0 for all n and (ii)  M columns span the task-adaptive classification space". The text make it sounds like your are finding the solution of (i) which happened to also verify (ii) which is not true. For instance, M=0 satistifies (i) but not (ii).

**Experience Assessment:**

I have read many papers in this area.

**Review Assessment: Checking Correctness Of Derivations And Theory:**

I carefully checked the derivations and theory.

**Review Assessment: Checking Correctness Of Experiments:**

I assessed the sensibility of the experiments.

**Review Assessment: Thoroughness In Paper Reading:**

I read the paper at least twice and used my best judgement in assessing the paper.

---

> ### Author Response · Authors · 2019-11-13
> **Response to Official Blind Review #3**
>
> Unfortunately, we must disagree with the reviewer in that our abstract and introduction very clearly state the goal, motivation and differentiating features of the proposed method. In particular, we make it clear that our method employs explicit task-conditioning in which unlabeled sample clustering for the current task takes place in a new projection space different from the embedding space. We also state that our method introduces a concept of controlling the degree of task-conditioning, the process in which the amount of task-conditioning varies with the number of updates for the projection space. We are not really giving the sketch of the algorithm but rather the key idea and the gist of the method. We do admit that there is a sentence in the abstract about soft labeling that could be viewed as a bit descriptive of the algorithm. Likewise, the second to the last paragraph of the introduction could be shortened. We have made these changes in our newly uploaded revision. Also, to help clarify the terminologies that have come from the work of TapNet, we have now provided a separate section on the background of TapNet. As for the impact of the proposed work, our message has been consistent in that iterative task-conditioning in utilizing unlabeled samples is a new notion that is significant and by doing so one could achieve state-of-the-art semi-supervised few-shot learning accuracies.
>
> We do not agree that our proposed algorithm is simply adding soft labeling to TapNet. The proposed repetitive updates of projection space provide an increasing degree of task-conditioning, which is a new concept, as already emphasized above. This argument is strengthened by the fact that the performance of the proposed method exceeds well beyond the simple combination of soft labeling and TapNet, especially for fewer-shot experiments. For example, for 1-shot miniImageNet, the simple combination gives a 52.82% accuracy, which improves to 55.50% with the proposed TAC. For 1-shot tieredImageNet, the improvement is from 54.18% to 58.48%.
>
> Clustering is a term not specific to our work but is rather a basic concept for semi-supervised learning. Classification space is also general; it should naturally mean a space where classification takes place. We have never used a term “per class network”; we used “per-class network output average”, which is a concept discussed in the well-known Prototypical Networks of Snell et al., 2017. As mentioned in our introduction, it is the per-class averages of the features that act as prototypes for representing given tasks.
>
> As for the construction/description of M, the reviewer seems to be under the impression that (i) and (ii) are two separate objectives in finding M. This is not true; as stated in the paper, any M that satisfies epsilon_n x M = 0 for all n does contain columns that span the task-adaptive classification space, unless of course M is all zero. Perhaps the reviewer is worrying about the trivial case of M being all zero. While this should be obvious, we have made it explicitly clear in the revised paper that we are barring the all-zero M when finding M.

---

### Decision · Program_Chairs · 2019-12-19

**Decision:**

Reject

**Comment:**

This paper proposes an approach to semi-supervised few-shot learning. In a discussion after the rebuttal phase, the reviewers were somewhat split on this paper, appreciating the advantages of the algorithm such as increased robustness to distractors and the ability to adapt with additional iterations, but were concerned that the contributions over Ren et al were not significant. Overall, the contributions of this paper don't quite warrant publication at ICLR.